# Novel magnetic $Fe_3O_4$/g-$C_3N_4$/$MoO_3$ nanocomposites with highly enhanced photocatalytic activities: Visible-light-driven degradation of tetracycline from aqueous environment

**Tianpei He[1], Yaohui Wu[1,2]\*, Chenyang Jiang[1], Zhifen Chen[1], Yonghong Wang[1], Gaoqiang Liu[1], Zhenggang Xu[2,3], Ge Ning[4], Xiaoyong Chen[1], Yunlin Zhao[2]**

**1** Hunan Provincial Key Laboratory for Forestry Biotechnology, Central South University of Forestry and Technology, Changsha, China, **2** Hunan Research Center of Engineering Technology for Utilization of Environmental and Resources Plant, Central South University of Forestry and Technology, Changsha, China, **3** Hunan Urban and Rural Ecological Planning and Restoration Engineering Research Center, Hunan City University, Hunan, China, **4** International Education Institute, Hunan University of Chinese Medicine, Changsha, China

\* wyh752100@163.com

**Data Availability Statement:** All the data supporting the ideas in this article are in the picture and table files.

## Abstract

In the present work, a series of magnetically separable $Fe_3O_4$/g-$C_3N_4$/$MoO_3$ nanocomposite catalysts were prepared. The as-prepared catalysts were characterized by XRD, EDX, TEM, FT-IR, UV-Vis DRS, TGA, PL, BET and VSM. The photocatalytic activity of photocatalytic materials was evaluated by catalytic degradation of tetracycline solution under visible light irradiation. Furthermore, the influences of weight percent of $MoO_3$ and scavengers of the reactive species on the degradation activity were investigated. The results showed that the $Fe_3O_4$/g-$C_3N_4$/$MoO_3$ (30%) nanocomposites exhibited highest removal ability for TC, 94% TC was removed during the treatment. Photocatalytic activity of $Fe_3O_4$/g-$C_3N_4$/$MoO_3$ (30%) was about 6.9, 5, and 19.9-fold higher than those of the $MoO_3$, g-$C_3N_4$, and $Fe_3O_4$/g-$C_3N_4$ samples, respectively. The excellent photocatalytic performance was mainly attributed to the Z-scheme structure formed between $MoO_3$ and g-$C_3N_4$, which enhanced the efficient separation of the electron-hole and sufficient utilization charge carriers for generating active radials. The highly improved activity was also partially beneficial from the increase in adsorption of the photocatalysts in visible range due to the combinaion of $Fe_3O_4$. Superoxide ions ($\cdot O_2^-$) was the primary reactive species for the photocatalytic degradation of TC, as degradation rate were decreased to 6% in solution containing benzoquinone (BQ). Data indicate that the novel $Fe_3O_4$/g-$C_3N_4$/$MoO_3$ was favorable for the degradation of high concentrations of tetracycline in water.

**Funding:** This work is supported in part by the Science and Technology Program of Changsha, China (kq1907097) and Central South University of Forestry and Technology Graduate Technology Innovation Fund (CX20190620).

**Competing interests:** The authors have declared that no competing interests exist.

## Introduction

Tetracycline (TC) has been widely used to treat bacterial infections in humans and animals over the past few decades [1]. Besides for medical applications, TCs are also employed as a supplement in animal husbandry to promote animal growth [2]. However, due to the widespread use of TC, TC residues could be frequently detected in various environmental matrices [3,4]. The residual TC in the environment would seriously threaten the ecosystem and public health [5]. In general, TC cannot be effectively removed by conventional wastewater treatment processes, such as biological treatment [6]. Therefore, new techniques are required to remove TC in water. Recently, photocatalytic assays have received a lot of intensive research interest worldwide due to its high efficiency and reliability, and have emerged as highly effective techniques for TC degradation from water [7]. Some photocatalysts have the function of degrading pollutants while Excellent antibacterial activity [8,9]. Common traditional photocatalysts, such as ZnO, $TiO_2$ [10], have been confirmed can degrade TC under light irradiation. However, in practical applications, these photocatalysts couldn't fully utilize solar energy, which causes them to be hindered in practical applications [11]. Therefore, the high efficient sunlight-driven photocatalysts are received lots of attention for the photocatalytic treatment of TC.

Graphitic carbon nitride (g-$C_3N_4$) has a strong visible-light response due to it easily produced electrons and holes under visible-light irritation [12]. Due to its advantages of low toxicity, low preparation cost and high stability, it has been applied to the removal of organic pollutants in water, which has aroused extensive research interest [13,14]. Unfortunately, g-$C_3N_4$ has low redox potential, and its photogenerated electron-hole pairs are easy to recombine [15]. These result in the limitation of its application as a self-sufficient semiconductor for the removal of contaminations by photodegradation [16]. Therefore, various methods have been evolved to enhance the photocatalytic activity of pure g-$C_3N_4$, including metal deposition [17,18], nonmetal doping [19], coupling with other materials [20], and using nano-sized structures [21]. By coupling g-$C_3N_4$ with other semiconductors to form a heterojunction structure, the shortcomings of high recombination rate of photogenerated electron-hole pairs of a single photocatalyst could be solved [22]. It should be noted that Z-scheme heterostructure formed by combining g-$C_3N_4$ with other semiconductors can efficiently separate the photogenerated electrons and holes, thereby improving the photocatalytic activity of g-$C_3N_4$ under visible light [23]. Yu et al., proposed a direct g-$C_3N_4$-$TiO_2$ Z-scheme photocatalyst, which increased the photocatalytic activity by 2.1 times compared to pure $TiO_2$ [24]. Hong et al., reported that the photocatalytic efficiency of a Z-scheme $V_2O_5$/ g-$C_3N_4$ heterojunction for the degradation RhB was as high as 7.3 and 13.0 times that of pure g-$C_3N_4$ and $V_2O_5$, respectively [25]. $MoO_3$ is a semiconducting material with wide gap, stable crystal structure, and photochromic sensitive nature [26]. It has been regarded as a promising candidate to form hybrid photocatalyst due to its special energetic and electrical properties [23]. Previous studies confirmed that combining with $MoO_3$, the photocatalitic activities of many photocatalyst, including $TiO_2$ [27], CdS [28], and polyimides [29], could be improved greatly. The composites possessed excellent photocatalytic activities by hindering charge recombination and improving charge transfer processes. Recently, researchers found that combining $MoO_3$ with g-$C_3N_4$ could produce Z-scheme photocatalyst. The photocatalytic performance was enhanced due to the suitable band gaps between the two semiconductors. Under light illumination, the photogenerated charge carrier can be efficiently separated and thus generated more reactive species [23,30]. However, most photocatalysts with high activity exist as nano-powders [31], and due to the small particle size, they can be easily dispersed in water thus couldn't be separated effectively [32]. This characteristic makes it practically limited and prone to secondary pollution [33]. To overcome the above problem, some magnetic materials, such as $Fe_3O_4$ and $CoFe_2O_4$, have been achieved

considerable attention [34,35]. Magnetic materials can transfer their magnetic properties to photocatalyst after being loaded, thus the photocatalyst can be separated effectively and easily from the treated solution using external magnetic field [36].

Spurred on by aforesaid information, after integration and envision, a novel ternary Z-scheme photocatalyst composites was presented combining $g$-$C_3N_4$ with $MoO_3$ and $Fe_3O_4$. The aim of this study was to develop an efficient photocatalyst by combining the interfacial connection of $g$-$C_3N_4$ and $MoO_3$ as well as the easy separation of magnetic materials. Their physical and chemical properties were investigated via a series of characterization. The TC-degrading ability of the prepared composites was studied. The influences of $MoO_3$ content on the photocatalytic performance of the composite were evaluated. The possible mechanisms for the photocatalytic activity enhancement and the TC degradation were presented.

## 1. Experimental

### 1.1. Material preparation

The $g$-$C_3N_4$ was prepared by direct heating of melamine to 520 $^o$C for 3 h in a muffle furnace, and the resultant samples was milled into powder for further use.

The $Fe_3O_4/g$-$C_3N_4$ was prepared by the following steps: $g$-$C_3N_4$ was dispersed in ethanol/water (1:2) solution and then treated with an ultrasonic cleaner at 300 W for 6h to form an uniform solution with 62.5 mg/L $g$-$C_3N_4$. 20 ml of 175 mg/L $FeCl_3$ and 20 ml of 68 mg/L $FeCl_2$ added into 500 mL of the suspension of $g$-$C_3N_4$. The mixture was stirred and dispersed at 80 $^o$C for 30 min prior to the quick injection of 10 mL of ammonia solution. The resultant mixture was stirred at 80 $^o$C for another 30 min. The as-obtained precipitate was washed several times with ultrapure water and absolute alcohol before being dried in air at 80 $^o$C for further use. The resultant sample was named $Fe_3O_4/g$-$C_3N_4$.

AHM (Ammonium heptamolybdate tetrahydrate) was added into ultrapure water with a little acetic acid. The resultant solution was adjusted to pH 3.5 with 36% acetic acid and stored at 80 $^o$C for 12 h to obtain amount of white precipitation. The precipitation was washed by absolute ethanol for 5 times and consequently dried in air at 60 $^o$C for 12 h (designed as secondary ammonium molybdate). After the obtained sample was ground for 30 minutes, it was sintered at 500 $^o$C for 2 hours under the protection of nitrogen. The resultant sample was named $MoO_3$.

The $Fe_3O_4/g$-$C_3N_4/MoO_3$ nanocomposites were synthesized by calcination method. Secondary ammonium molybdate and $Fe_3O_4/g$-$C_3N_4$ were taken separately in mortars, grounded for 30 mins. Then the two samples were mixed and thoroughly grounded for another 30 mins before being sintered at 500 $^o$C for 2 h under nitrogen atmosphere. After being cooled, the product was obtained. Following the same synthesis route different weight percentage of $Fe_3O_4/g$-$C_3N_4/MoO_3$ nanocomposites were obtained varying the wt% of secondary ammonium molybdate maintaining wt ratio 10, 20, 30 and 40 wt%. All the $Fe_3O_4/g$-$C_3N_4/MoO_3$ composites were denoted as $Fe_3O_4/g$-$C_3N_4/MoO_3$(10%), $Fe_3O_4/g$-$C_3N_4/MoO_3$(20%), $Fe_3O_4/g$-$C_3N_4/MoO_3$(30%), and $Fe_3O_4/g$-$C_3N_4/MoO_3$(40%).

### 1.2. Characterization

The XRD patterns were obtained by a Bruker D8 Advance X-ray diffractometer with CuKα radiation, employing scanning rate of 0.02$^o$/sec in the 2θ range from 5$^o$ to 90$^o$. Surface morphology was studied by JSM-7500F SEM, using an accelerating voltage of 5 kV. The purity and elemental analysis of the products were obtained by EDX on JSM-7500F SEM. The microstructures were investigated by a JEM-2100F TEM with an acceleration voltage of 200 kV. HRTEM was conducted on a JEM-2100F. The UV-Vis DRS was performed by an UV270

spectrophotometer, utilizing $BaSO_4$ was the reflectance. The FT-IR spectra were studied by a Nicolet-iS10 instruction. XPS data was obtained by an Escalab 250Xi apparatus. The surface area and pore properties were estimated by BET and BJH models using the adsorption data collected by Micro for TriStar II Plus 2.02 apparatus at -196 $^o$C. Thermo-gravimetric analysis (TGA) was carried out on a STA 449F3 thermal analyzer with a heating rate of 10 $^o$C/min from room temperature to 1000 $^o$C in an air flow. The photoluminescence (PL) spectra were obtained by a Fls980 fluorescence spectrophotometer with an excitation wavelength of 380 nm. Magnetic properties were investigated using a MPMS.

### 1.3. Photocatalytic activity measurement

The capacity of the synthesized catalysts to photodegrade TC was performed by a photochemistry reaction instrument (YM-GHX-V, Shanghai Yuming Instrument Co. Ltd, China) with a 1000 W Xe lamp applied as visible light source, as shown in S1 Fig. In the reaction system, the reaction solution is packed in a quartz tube with a capacity of 50 ml, and the quartz tube is fixed at a distance of 2 cm from the light source. An optical power meter (OPT-1A, China) was used to measure the intensity of the experimental lamp to be 37.5 mW/cm$^2$ ($\lambda$ >400 nm). A water circulation system was utilized to keep the reaction system at 15 $^o$C. In each experiment, 10 mg of the photocatalyst was added into 50 mL of TC solution (40 mg/L). Prior to illumination, the reaction solution was treated in dark for 30 min to achieve adsorption-desorption equilibium. Every 30 minutes, 0.5 mL was sampled from the reaction solution and centrifuged immediately at 5000 rpm for 7 min. The TC concentration was determined based on absorbance at 355 nm by Nano Drop 2000 spectrophotometer.

### 1.4. Active species trapping measurement

Radical scavenge experiments was performed to verify the role of active substances in the degradation of TC. Ethylenediaminetetraacetate (EDTA-2Na, 1 mM), potassium dichromate ($K_2Cr_2O_7$, 50 $\mu$M), isopropanol (IPA, 10 mM), and benzoquinone (BQ, 1 mM), were respectively applied as the trapping agent of $h^+$, $e^-$, $\cdot OH$, and $\cdot O_2^-$ [37,38].

## 2. Results and discussion

### 2.1. Photocatalyst characterization

Fig 1 showed the typical XRD patterns of $MoO_3$, $g$-$C_3N_4$, $Fe_3O_4$, $Fe_3O_4$/$g$-$C_3N_4$ and $Fe_3O_4$/$g$-$C_3N_4$/$MoO_3$ composites. It could clearly observed that the (020), (110), (040), (021), (111), (060), and (200) peaks of $MoO_3$ were at 12.83$^o$, 23.46$^o$, 25.76$^o$, 27.40$^o$, 33.75$^o$, 39.07$^o$, and 46.04$^o$, which could be exactly indexed as the orthorhombic structure ($\alpha$-$MoO_3$) (JCPDF 35–0609) [39]. Previous study reported that $MoO_3$ had three different crystalline structure, orthorhombic ($\alpha$-$MoO_3$), monoclinic ($\beta$-$MoO_3$) and hexagonal (h-$MoO_3$) and $\alpha$-$MoO_3$ was thermodynamically stable [40]. So it concluded that the proposed synthesis process benefit the growth of $\alpha$-$MoO_3$ which was more thermodynamically stable than $\beta$-$MoO_3$. The (100) and (002) peaks of $g$-$C_3N_4$ appeared at 13.12$^o$ and 27.52$^o$, which were in consistent with the characteristic interplanar staking peaks of the inter-layer structural packing and aromatic systems, respectively [41]. The main peaks of $Fe_3O_4$ appeared at 35.83$^o$, 43.18$^o$, 53.17$^o$, 57.43$^o$ and 63.04$^o$, well presented to the lattice plane (311), (400), (422), (511) and (440), respectively [42]. The $Fe_3O_4$/$g$-$C_3N_4$ nanocomposites had the peaks corresponding $Fe_3O_4$ and $g$-$C_3N_4$, indicting $Fe_3O_4$ were successfully deposited on $g$-$C_3N_4$ surface. The patterns for $Fe_3O_4$/$g$-$C_3N_4$/$MoO_3$ nanocomposites were composed of the diffraction peaks corresponding to $g$-$C_3N_4$, $MoO_3$ and $Fe_3O_4$, confirming the coexistence of the three materials. Moreover, it was clearly that the

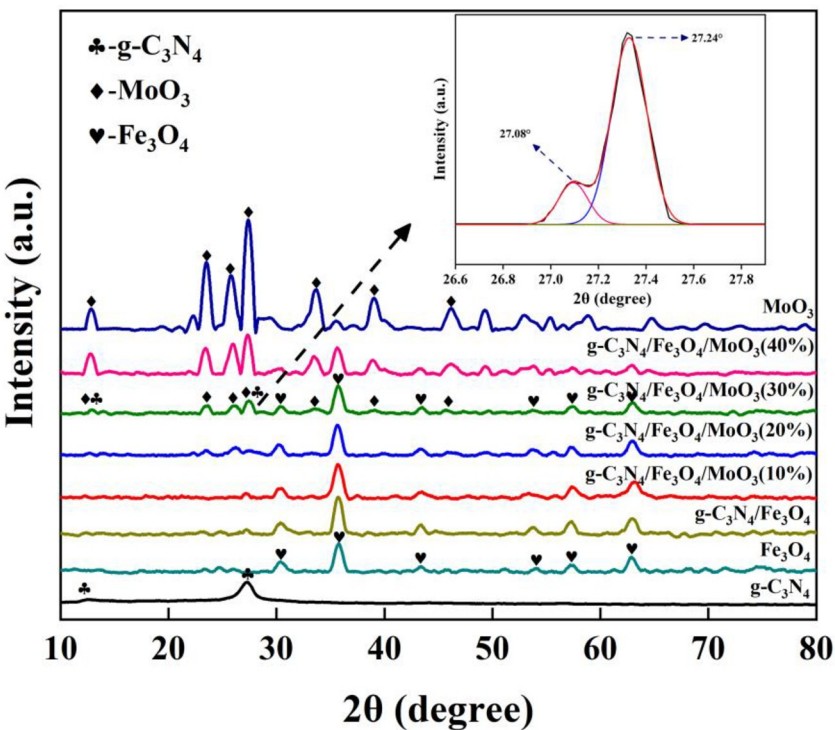

**Fig 1. XRD patterns for the MoO₃, g-C₃N₄, Fe₃O₄, Fe₃O₄/g-C₃N₄ and Fe₃O₄/g-C₃N₄/MoO₃ nanocomposites.** Inset image shows the deconvulation peaks for MoO₃ and g-C₃N₄.

intensity of the peaks for $MoO_3$ in $Fe_3O_4$/g-$C_3N_4$/$MoO_3$ nanocomposites increased with the increase of the weight percent of $MoO_3$. However, the peaks for g-$C_3N_4$ in the nanocomposites were not obviously observed as it overlapped with the peaks for $MoO_3$. The inset XRD patters for $Fe_3O_4$/g-$C_3N_4$/$MoO_3$ (30%) displayed the two deconvulation peaks at 27.32, suggesting the presence of both $MoO_3$ and g-$C_3N_4$. These results further verified that $Fe_3O_4$ and g-$C_3N_4$ combined with $MoO_3$ successfully.

Fig 2 exhibited the elemental mapping of the $Fe_3O_4$/g-$C_3N_4$/$MoO_3$ (30%) nanocomposites which was detected from a randomly selected area of the nanocomposite using EDX detector. It could be clearly found C, N, Fe, O and Mo (Fig 2B–2F) were all homogeneous indicating uniform distributions of $Fe_3O_4$, g-$C_3N_4$, and $MoO_3$ in the selected area of the corresponding SEM image (Fig 2A).

Fig 3 presented the morphology and microstructure of the $MoO_3$, g-$C_3N_4$, $Fe_3O_4$/g-$C_3N_4$, and $Fe_3O_4$/g-$C_3N_4$/$MoO_3$ (30%) samples investigated by TEM. It was obviously that $MoO_3$ possessed Flake-like structure with the size of about 200 nm (Fig 3(A)). Pure g-$C_3N_4$ (Fig 3(B)) shows lamellar-like and smooth morphology. In $Fe_3O_4$/g-$C_3N_4$ composites (Fig 3(C)), dark $Fe_3O_4$ nanoparticles with a particle size of 10–20 nm were deposited on the surface. For $Fe_3O_4$/g-$C_3N_4$/$MoO_3$ (30%) composites (Fig 3(D)), the composites of $Fe_3O_4$/g-$C_3N_4$ were well adhered on the surface of $MoO_3$. These results demonstrated the successful synthesis of the ternary $Fe_3O_4$/g-$C_3N_4$/$MoO_3$. To further verify the formation of $Fe_3O_4$/g-$C_3N_4$/$MoO_3$ ternary structure, HRTEM image it has been used to investigate the microstructure of 30% $Fe_3O_4$/g-$C_3N_4$/$MoO_3$ (Fig 3(E)). The HRTEM image illustrated that the heterostructure of $Fe_3O_4$/ g-$C_3N_4$/ $MoO_3$ composite material showed lattice fringes of 0.38 nm corresponded to the (110) plane of $MoO_3$, the fringes of 0.29 nm assigned to the (220) plane of $Fe_3O_4$. The interaction

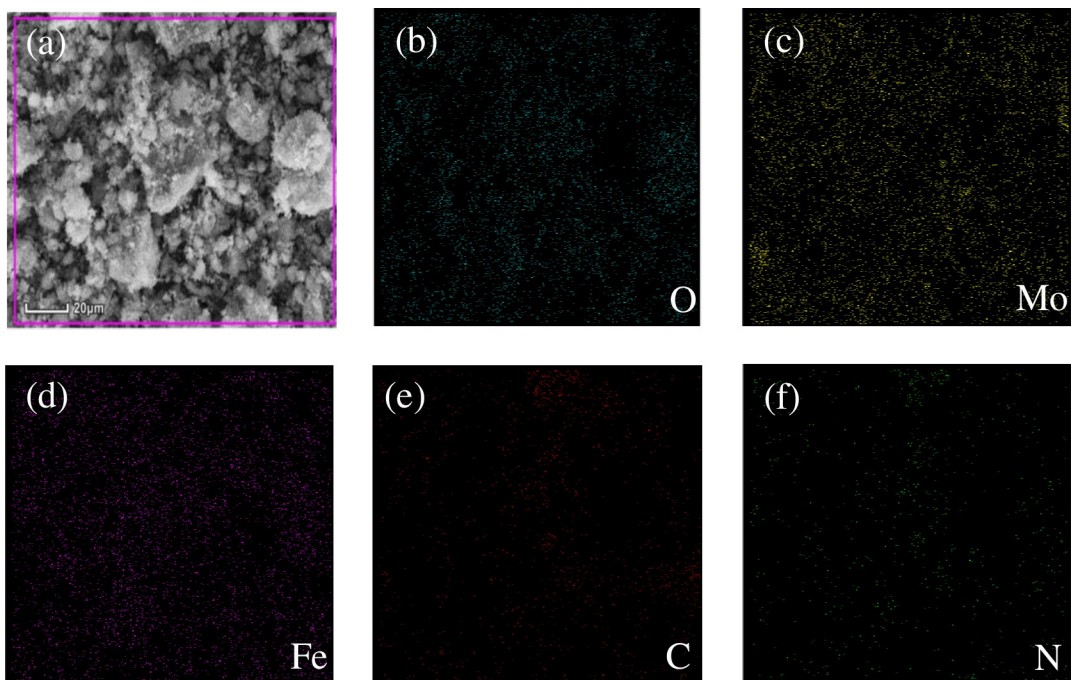

**Fig 2.** (a) SEM images of Fe$_3$O$_4$/g-C$_3$N$_4$/MoO$_3$ (30%); (b-f) EDX mapping for the Fe$_3$O$_4$/g-C$_3$N$_4$/MoO$_3$ (30%) nanocomposite.

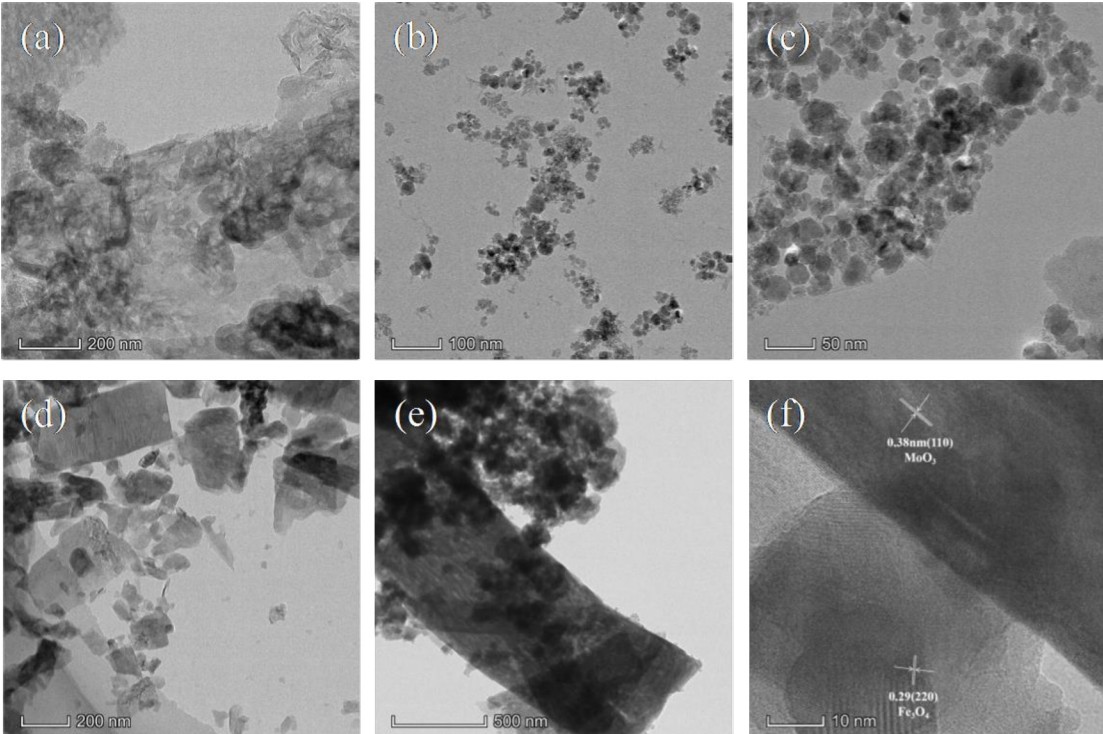

**Fig 3.** TEM of (a) g-C$_3$N$_4$; (b) Fe$_3$O$_4$; (c) Fe$_3$O$_4$/g-C$_3$N$_4$; (d) MoO$_3$; (e) Fe$_3$O$_4$/g-C$_3$N$_4$/MoO$_3$ (30%);(f) HRTEM images of the Fe$_3$O$_4$/g-C$_3$N$_4$/MoO$_3$ (30%) nanocomposite.

between $MoO_3$ and g-$C_3N_4$ could benefit a continuous flow of electrons between them due to the improvement of electron channelization through the interface [43], resulting in the improvement of photocatalytic efficiency.

X-ray photoelectron spectroscopy (XPS) was carried out to further analyze the surface compositions and chemical states of $Fe_3O_4$/g-$C_3N_4$/$MoO_3$ (30%) sample (Fig 4). Fig 4(A) revealed the presence of Mo, C, N, O and Fe elements on the surface of the as-prepared sample. The photoelectron lines at binding energy of 233, 285, 363, 399, 531 and 712 eV were correspond to Mo 3d, C 1s, N 1s, O 1s and Fe 2p in the sample, respectively [44]. Fig 4(B) represented the XPS spectrum of Fe. The two Fe $2p_{3/2}$ and $2p_{1/2}$ peaks corresponding to binding energy 710.6 and 723.7 eV without shakeup satellite peak of $Fe_2O_3$, and their binding energy was consistent with that in pure $Fe_3O_4$ [45], suggesting the coexistence of dual iron oxidation states of $Fe^{2+}$ and $Fe^{3+}$ [46]. The binding energy spectrum of Mo was demonstrated in Fig 4(C), there were only two peaks existed at 232.1 and 235.3 eV corresponding to $3d_{5/2}$ and $3d_{3/2}$ of Mo atom in +6 oxidation states [47]. The C 1s signal could be divided into four peaks at 284.2, 285.8, 288.1, and 289.5 eV, implying the presence of chemically different carbon species in the sample (Fig 4(D)). The peaks located at 284.2 and 285.8 were attributed to C = C and C-O bonds, respectively [48]. The peak located at 288.1 eV was attributed to $sp^2$ hybridised C atoms in the triazine rings inside thearomatic structure, while the peak at 289.5 eV was corresponded to N = C-N group or -$NH_2$ group as originating from g-$C_3N_4$. The XPS peak of N 1s (Fig 4(E)) obviously centered at the binding energy of 398.0 eV, which could be assigned to the $sp^2$ hybridized nitrogen (C = N-C) whereas peak at and 401.2 eV represented the tertiary nitrogen (N-$C_3$). Based on Fig 4(F), there were two types of oxygen species, which should assign to the O 1s peak. The offering of the anionic oxygen in $Fe_3O_4$ centered at about 530.1 eV, and the oxygen in $MoO_3$ centered at 531.7 eV [49]. The XPS results strongly suggested the coexistence of $Fe_3O_4$, g-$C_3N_4$, and $MoO_3$.

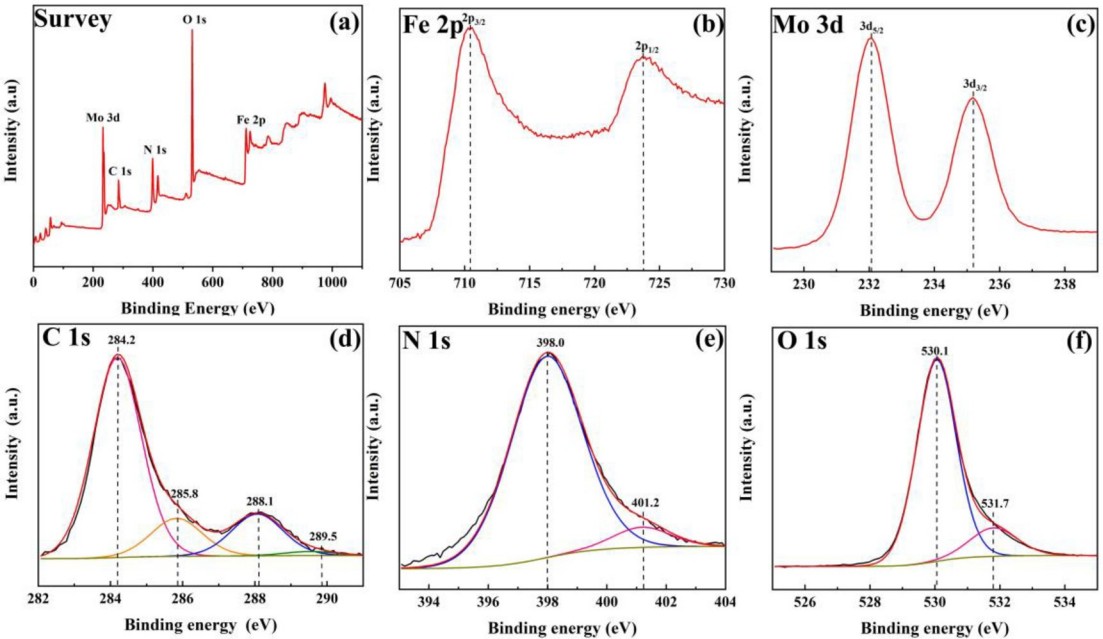

**Fig 4. XPS patterns of $Fe_3O_4$/g-$C_3N_4$/$MoO_3$ (30%) nanocomposite.** (a) Survey spectra; (b)Fe 2p; (c) Mo 3d; (d) C 1s; (e) N 1s; (f) O 1s.

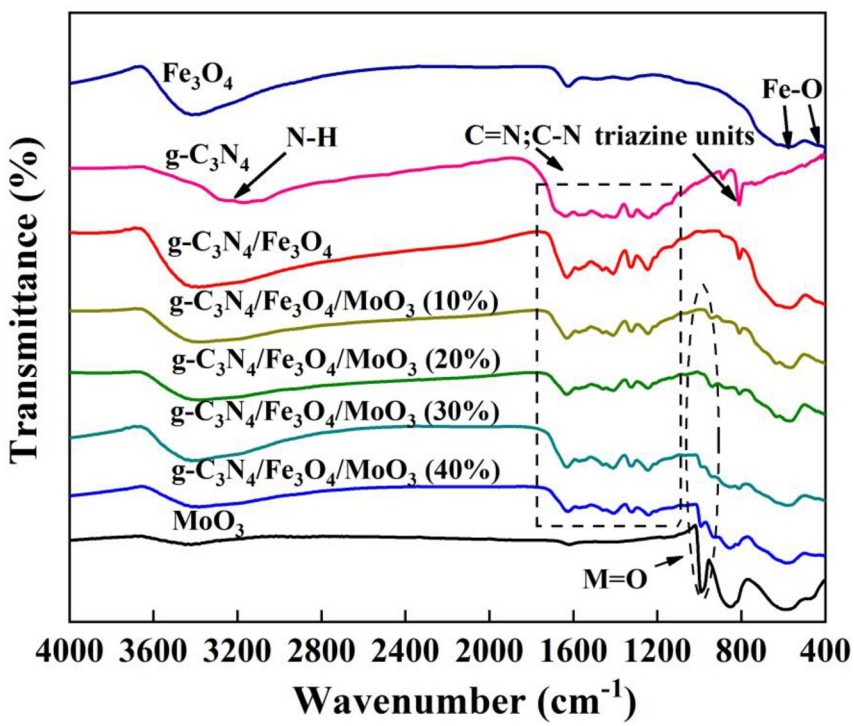

**Fig 5. FT-IR spectra of MoO$_3$, g-C$_3$N$_4$, Fe$_3$O$_4$, Fe$_3$O$_4$/g-C$_3$N$_4$ and Fe$_3$O$_4$/g-C$_3$N$_4$/MoO$_3$ nanocomposites.**

Chemical structures of the MoO$_3$, g-C$_3$N$_4$, Fe$_3$O$_4$, Fe$_3$O$_4$/g-C$_3$N$_4$ and various Fe$_3$O$_4$/g-C$_3$N$_4$/MoO$_3$ nanocomposites were studied by FT-IR spectra, and the results were exhibited in Fig 5. For pure g-C$_3$N$_4$, the absorption signal of 3165 cm$^{-1}$ was beneficial from the stretching vibrations of N-H. The strong absorption band in the range of 1240–1650 cm$^{-1}$ is correspond to typical skeletal stretching vibrations of C-N and C = N in s-triazine or tri-striazine [50]. Simultaneously, the band at 809 cm$^{-1}$ can be assigned to the typical breathing mode of the heptazine arrangement [51]. In case of pure Fe$_3$O$_4$ nanoparticles, two peaks at 566 and 421 cm$^{-1}$ were corresponded to the stretching vibrations of Fe-O [52]. Neat MoO$_3$ showed signals of 599 cm$^{-1}$ and 852 cm$^{-1}$ which were related to the stretching vibrational modes of O shared by three Mo and the Mo-O-Mo unit respectively in the crystalline α-MoO$_3$. In addition, a signal at 991 cm$^{-1}$ was due to Mo = O for the crystalline α-MoO$_3$ [53]. In the Fe$_3$O$_4$/g-C$_3$N$_4$/MoO$_3$ nanocomposites, the existence of the typical vibrational modes of g-C$_3$N$_4$, Fe$_3$O$_4$, and MoO$_3$ indicated the coexistence of these three contents in the nanocomposites.

S2 Fig displays TGA curves for the g-C$_3$N$_4$ and Fe$_3$O$_4$/g-C$_3$N$_4$/MoO$_3$ (30%) samples. As can be seen, the pristine g-C$_3$N$_4$ shows a weight loss of about 96% after heating up to 750 $^{\circ}$C. Hence, it was concluded that the g-C$_3$N$_4$ decomposes almost completely heating up to 750 $^{\circ}$C. It is evident that the thermal behavior of Fe$_3$O$_4$/g-C$_3$N$_4$ and Fe$_3$O$_4$/g-C$_3$N$_4$/MoO$_3$ (30%) samples are similar to that of g-C$_3$N$_4$. As can be seen, by loading Fe$_3$O$_4$ and MoO$_3$ on the g-C$_3$N$_4$ sheets, thermal degradation of the nano-composites starts from lower temperature relative to the pristine g-C$_3$N$_4$. Hence, similar to many g-C$_3$N$_4$-based nanocomposites, thermal stability of the pristine g-C$_3$N$_4$ decreases with depositing different particles [45,46]. The g-C$_3$N$_4$ contents of Fe$_3$O$_4$/g-C$_3$N$_4$ and Fe$_3$O$_4$/g-C$_3$N$_4$/MoO$_3$ (30%) nanocomposites were calculated from the weights remaining after heating the samples to over 650 $^{\circ}$C. The g-C$_3$N$_4$ contents of the Fe$_3$O$_4$/g-C$_3$N$_4$/MoO$_3$ (30%) nanocomposite was about 8.2%, respectively. As can be seen, besides the weight loss of g-C$_3$N$_4$, another weight loss between 750 and 1000 $^{\circ}$C in the Fe$_3$O$_4$/

**Table 1. Weight percentages of different compounds in the Fe₃O₄/g-C₃N₄/MoO₃ 30%) nanocomposite.**

| Compound | Weight percentage |
|---|---|
| g-C₃N₄ | 8.2 |
| MoO₃ | 16.4 |
| Fe₃O₄ | 75.4 |

g-C₃N₄/MoO₃ (30%) composites, could be ascribed to the vaporization of MoO₃. The MoO₃ contents of the Fe₃O₄/g-C₃N₄/MoO₃ (30%) is about 16.4%. In addition, after calculation, The MoO₃ contents of the Fe₃O₄/g-C₃N₄/MoO₃ (30%) is about 75.4%. The results were listed in Table 1.

It was well known that the photoabsorptive capacity of a photocatalyst would greatly affect its photocatalytic activity [54]. Thus, UV-Vis DRS was used to investigate the photoabsorption ability of a series of as-prepared samples and the results were showed in Fig 6. As could be seen in Fig 6(A), both pristine g-C₃N₄ and MoO₃ possessed small absorption in visible region and had absorption edges at about 470 nm, which were compatible with the reported absorption edges for g-C₃N₄ and MoO₃ [55]. Fig 6(B) displayed the band gaps of g-C₃N₄ and MoO₃ were consistent with previous studies, which were 2.72 eV and 2.85 eV, respectively [20]. The band gap of all as-prepared photocatalysts were obtained by using Tauc's equation (Eq 1).

$$\alpha h v = A(h v - Eg)^{n/2} \tag{1}$$

where, $\alpha$, $h$, $v$, and A were absorption coefficient, Planck's constant (eV. s), the light frequency ($s^{-1}$), and proportionality constant, respectively. $Eg$ was the band gap, and $n$ was the power which was assumed to be 1 and 4 for direct and indirect transitions, respectively [56,57]. As displayed in the figure, the addition of Fe₃O₄ to the pure g-C₃N₄ greatly enhanced the absorption in the visible range. Interesting, the addition of MoO₃ to the Fe₃O₄/g-C₃N₄ slightly decreased the visible light absorption when the weight percentages of MoO₃ were lower than 30%. The absorption would be significantly reduced when the content of MoO₃ was over the value. However, compared to pristine g-C₃N₄ and MoO₃, the visible light absorption of Fe₃O₄/g-C₃N₄/MoO₃ nanocomposites was considerably high. These facts possibly make Fe₃O₄/g-

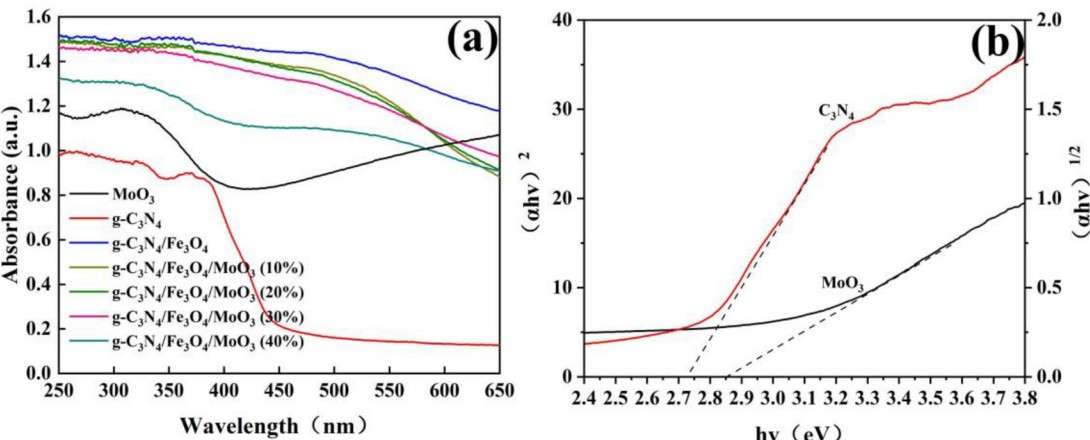

**Fig 6.** (a) UV-Vis diffuse reflectance absorption spectra for the MoO₃, g-C₃N₄, Fe₃O₄, Fe₃O₄/g-C₃N₄ and Fe₃O₄/g-C₃N₄/MoO₃ nanocomposites; (b) The corresponding calculated band gaps of g-C₃N₄ and MoO₃.

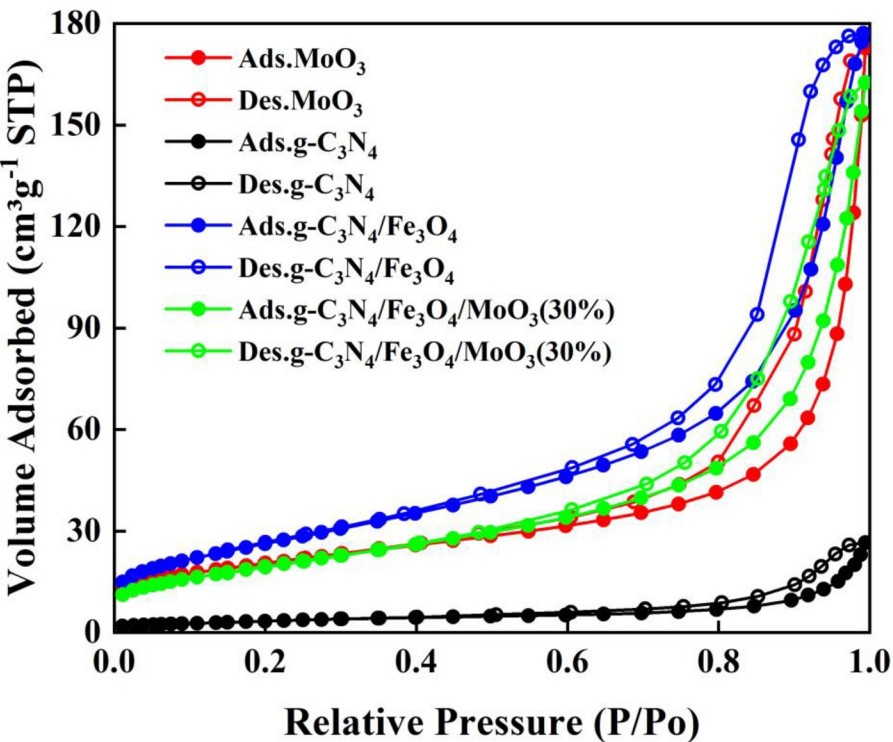

**Fig 7. Nitrogen adsorption-desorption isotherms of $MoO_3$, $g-C_3N_4$, $Fe_3O_4$, $Fe_3O_4/g-C_3N_4$ and $Fe_3O_4/g-C_3N_4/$ $MoO_3$ (30%) nanocomposite.**

$C_3N_4/MoO_3$ to use more visible light, and produce more photoexcited charge carriers than pure $g-C_3N_4$ or $MoO_3$.

To analyze textural properties of the prepared $MoO_3$, $g-C_3N_4$, $Fe_3O_4/g-C_3N_4$, and $Fe_3O_4/g-$ $C_3N_4/MoO_3$ (30%) photocatalysts, the results about $N_2$ adsorption-desorption isotherm were provided in Fig 7. As could be seen, the isotherm of each sample was of typical IV with $H_3$ hysteresis, indicating a characteristic of mesopores structure [58], which benefited to decreasing mass transfer limitations and harvesting light in the photocatalytic process [59]. BET and BJH models were used to investigate the specific surface areas and pore features of the four photocatalysts, respectively and the results were presented in Table 2. The surface areas of the $MoO_3$, $g-C_3N_4$, $Fe_3O_4/g-C_3N_4$, and $Fe_3O_4/g-C_3N_4/MoO_3$ (30%) were 73.1, 12.6, 97.4, and 72.7 $m^2g^{-1}$, respectively. Compared to single-phase $g-C_3N_4$, $Fe_3O_4/g-C_3N_4$ had larger surface area, which might attribute to the formation of hierarchical structure after loading $Fe_3O_4$ on $g-C_3N_4$ [60]. However, after the $Fe_3O_4/g-C_3N_4$ being modifying with $MoO_3$, the surface area was decreased. This decrease probably caused by the covering of the $Fe_3O_4/g-C_3N_4$ surface by $MoO_3$, resulting in the blocking of some active sites on the surface [61]. Generally, a decreased in the specific

**Table 2. The textural properties of $g-C_3N_4$, $Fe_3O_4/g-C_3N_4$, $Fe_3O_4/g-C_3N_4/MoO_3$ (30%) samples.**

| Photocatalyst | Surface area ($m^2 g^{-1}$) | Mean pore diameter (nm) | Total pore volume ($cm^3 g^{-1}$) |
|---|---|---|---|
| $MoO_3$ | 73.0615 | 14.63057 | 0.267233 |
| $g-C_3N_4$ | 12.6271 | 13.03849 | 0.041159 |
| $Fe_3O_4/g-C_3N_4$ | 97.4179 | 11.24809 | 0.273941 |
| $Fe_3O_4/g-C_3N_4/MoO_3$ (30%) | 72.6855 | 13.84626 | 0.251606 |

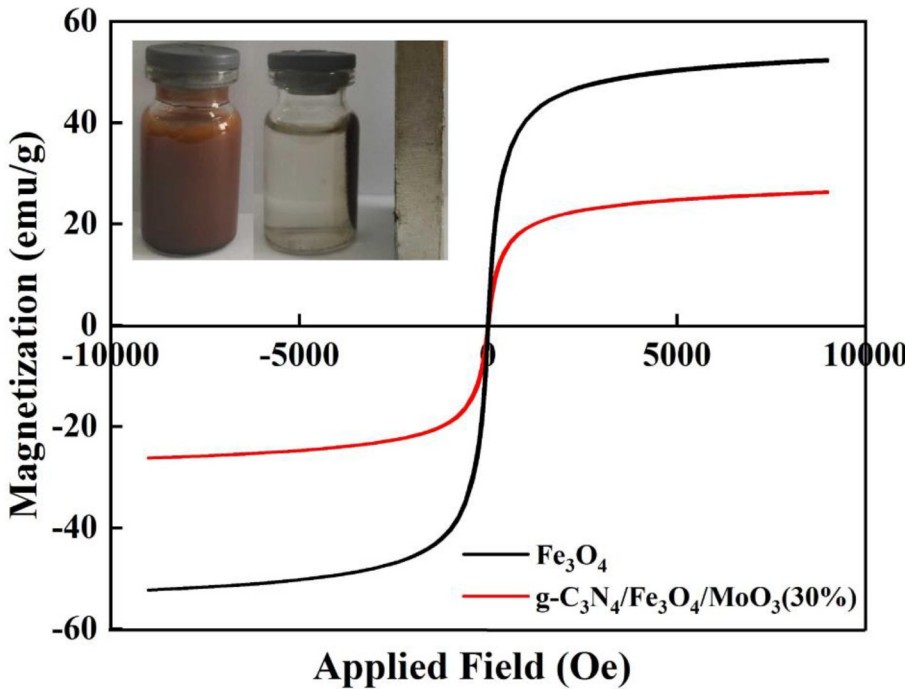

**Fig 8. Magnetization curves for the Fe₃O₄ nanoparticles and Fe₃O₄/g-C₃N₄/MoO₃ (30%) nanocomposite.** Inset of the figure shows separation of the nanocomposite from the treated solution using a magnet.

surface area of a semiconductor was accompanied by a decrease in its photocatalytic activity. Hence, the highly improved photocatalytic activity of $Fe_3O_4$/g-$C_3N_4$/$MoO_3$ (30%) should be not described to its textural properties.

Fig 8 displayed the VSM curves for the $Fe_3O_4$ nanoparticles and $Fe_3O_4$/g-$C_3N_4$/$MoO_3$ (30%) photocatalyst at ambient temperature. Saturation magnetization of the $Fe_3O_4$ nanoparticles was 52.5 emu/g, while that of the $Fe_3O_4$/g-$C_3N_4$/$MoO_3$ (30%) nanocomposites decreased to 26.3 emu/g due to the existence of the non-magnetic g-$C_3N_4$ and $MoO_3$. However, both of the samples displayed super paramagnetic behavior. By pacing an external magnet beside the glass bottle containing the $Fe_3O_4$/g-$C_3N_4$/$MoO_3$ (30%) nanocomposite, the particles were rapidly attracted to the wall of the glass bottle, as shown in the top-left inset of Fig 8, suggesting an easy separation under external magnetic field.

## 2.2. Photocatalytic activity and mechanism

Degradation of TC solution under visible light to evaluate the photocatalytic activity of the as-prepared catalyst, and the results were demonstrated in Fig 9. As shown in Fig 9(A), the blank experiments (in absence of any photocatalyst) revealed that the changes of TC concentration were negligible, that mean TC was quite stable under light irradiation, thus the self-degradation of TC was ruled out. The removal percentage of TC was denoted as $C/C_0$, in which $C$ was the TC concentration after adsorption and light illumination for a certain time, and $C_0$ was the initial concentration of TC. For pristine $MoO_3$, there were 20% TC were adsorbed and only about 17% TC were photodegraded in 120 min. Single-phase g-$C_3N_4$ displayed almost no adsorption and moderate photocatalytic activity for TC, with a removal percentage of 28% after 120 min under visible light. It should be noted that, when the $Fe_3O_4$ loaded on g-$C_3N_4$, the photodegrading ability decreased, with a removal percentage of 10%, implying $Fe_3O_4$ had a

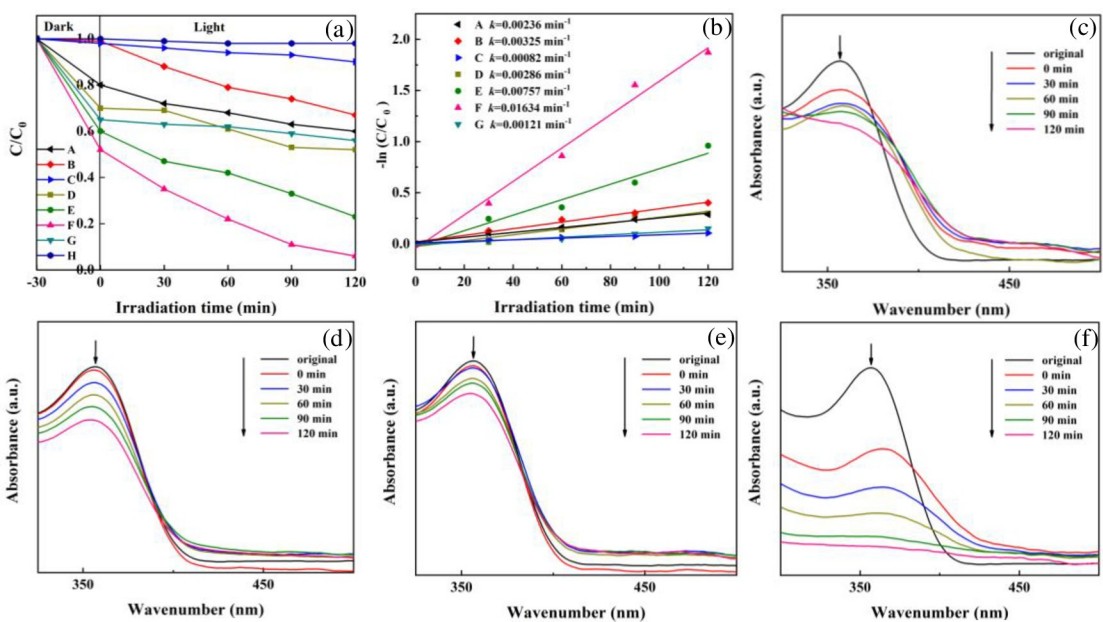

**Fig 9.** (a) Comparison of the photocatalytic activities of different samples. (A) MoO$_3$; (B) g-C$_3$N$_4$; (C) Fe$_3$O$_4$/g-C$_3$N$_4$; (D) Fe$_3$O$_4$/g-C$_3$N$_4$/MoO$_3$ (10%); (E) Fe$_3$O$_4$/g-C$_3$N$_4$/MoO$_3$ (20%); (F) Fe$_3$O$_4$/g-C$_3$N$_4$/MoO$_3$ (30%); (G)Fe$_3$O$_4$/g-C$_3$N$_4$/MoO$_3$ (40%) (H)No photocatalyst; (b) Pseudo-first-order kinetic curves of the corresponding samples; (c), (d), (e),and (f) Temporal evolutions of the spectra during the photocatalytic degradation of TC over MoO$_3$, g-C$_3$N$_4$, Fe$_3$O$_4$/g-C$_3$N$_4$,and Fe$_3$O$_4$/g-C$_3$N$_4$/MoO$_3$ (30%).

negative effect on the photocatalytic activities. The incorporation of the MoO$_3$ boosted the overall activity, and the Fe$_3$O$_4$/g-C$_3$N$_4$/MoO$_3$ composites displayed remarkable enhancements in the photodegrading-abilities. After irradiated for 120 min, the removal percentages were about 48, 77, 94, and 44% for 10, 20, 30, and 40% Fe$_3$O$_4$/g-C$_3$N$_4$/MoO$_3$ nanocomposites, in which 18, 37, 46, and 9% were attributed to photodegradation, respectively. Obviously, the nanocomposites with 30% MoO$_3$ possessed the best photocatalytic activity. Since the Fe$_3$O$_4$ had no positive effect on the photodegrading-ability, the improvements in the photacatalytic performance of the nanocomposites should attribute to the cooperation of g-C$_3$N$_4$ and MoO$_3$. Furthermore, when the weight percent of MoO$_3$ was over 30%, the degradation of TC decreased. That was to say, excess load of MoO$_3$ leaded to the lower photodegradating-ability, which implied that the superfluous MoO$_3$ could impede the interaction of g-C$_3$N$_4$ and MoO$_3$. The pseudo-first-order kinetic model ($\ln[TC] = \ln[TC]_0 - k_{obs}t$) was used to fit with the degradation process to quantify the activities of the resultant samples, in which the value of the observed first-order rate constant ($k_{obs}$) was equal to the corresponding slope of the straight line [62]. As shown in Fig 9(B), The $k$ of MoO$_3$, g-C$_3$N$_4$, Fe$_3$O$_4$/g-C$_3$N$_4$, and Fe$_3$O$_4$/g-C$_3$N$_4$/MoO$_3$ (30%) nanocompositeis were $2.36\times10^{-3}$, $3.25\times10^{-3}$, $8.2\times10^{-4}$, and $1.63\times10^{-2}$ min$^{-1}$, respectively. Thus, it could be concluded that activity of the Fe$_3$O$_4$/g-C$_3$N$_4$/MoO$_3$ (30%) nanocompositeis was about 6.9, 5 and 19.9-fold higher than those of MoO$_3$, g-C$_3$N$_4$, and Fe$_3$O$_4$/g-C$_3$N$_4$ composites, respectively. Fig 9(C)–9(F) displayed the UV-Vis spectral variation of TC solution during the adsorption and photodegradation over the MoO$_3$, g-C$_3$N$_4$, Fe$_3$O$_4$/g-C$_3$N$_4$, and Fe$_3$O$_4$/g-C$_3$N$_4$/MoO$_3$ (30%) nanocompositeis. For all the samples, the maximal absorbance at 355 nm decreased as the reaction progressed, suggesting gradual removal of TC. Comparison of Fe$_3$O$_4$/g-C$_3$N$_4$/MoO$_3$ (30%) with other similar reported systems of Fe$_3$O$_4$/g-C$_3$N$_4$ composites has been discussed in S1 Table.

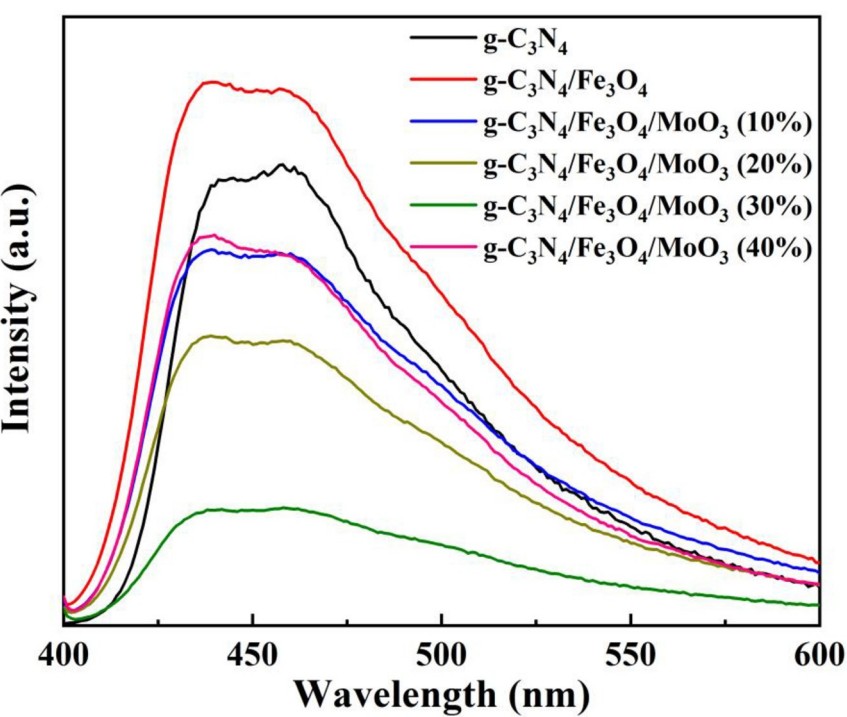

**Fig 10. PL spectra of g-C$_3$N$_4$, Fe$_3$O$_4$, Fe$_3$O$_4$/g-C$_3$N$_4$ and Fe$_3$O$_4$/g-C$_3$N$_4$/MoO$_3$ nanocomposites.**

In generally, for most semiconductors the photo-induced e$^-$/h$^+$ pairs can recombine after being irradiated by light thus emit fluorescence, which can be indicated by PL. Higher PL intensity of a semiconductor indicates a higher recombination rate of its e$^-$/h$^+$ pair [63]. Fig 10, showed the PL spectroscopy of g-C$_3$N$_4$, Fe$_3$O$_4$/g-C$_3$N$_4$, and Fe$_3$O$_4$/g-C$_3$N$_4$/MoO$_3$ series samples. As seen in the figure, g-C$_3$N$_4$ displayed large PL signal due to the high recombination of photo-induced e$^-$/h$^+$ pairs and low quantum yield [16]. However, Fe$_3$O$_4$/g-C$_3$N$_4$ nanocomposites exhibited a stronger PL than that of the pure g-C$_3$N$_4$, indicating a lower separation rate of photo-induced e$^-$/h$^+$ pairs. Interestingly, the addition of MoO$_3$ to the Fe$_3$O$_4$/g-C$_3$N$_4$ nanocomposites followed by the formation of the Fe$_3$O$_4$/g-C$_3$N$_4$/MoO$_3$ obviously reduced the PL emission intensity due to the combination of MoO$_3$ and Fe$_3$O$_4$/g-C$_3$N$_4$, which suggested the fabrication of the nanocomposites efficiently enhanced the of separation of e$^-$/h$^+$ pairs on the surface. It should be noted that the PL signal increased significantly when the content of MoO$_3$ were over 30%, implying an easier recombination of photogenerated charge carriers. The incensement may attribute to the agglomeration of the overloaded MoO$_3$ on the surface of the nanocomposites, resulting in the reduction of the interface area between g-C$_3$N$_4$ and MoO$_3$.

It had been reported that the ·O$_2^-$, ·OH, e$^-$ and h$^+$ were the main active species attributed to the photodegradation contaminants during the photocatalytic reactions [64]. However, their contribution to the degradation of contaminants was not identical and could be investigated by utilizing the quenching experiments. In order to estimate the role of each radical in the TC photodegradation, EDTA-2Na, K$_2$Cr$_2$O$_7$, IPA, and BQ were respectively used as the quenchers for h$^+$, e$^-$, ·OH, and ·O$_2^-$ in the TC degradation process in the Fe$_3$O$_4$/g-C$_3$N$_4$/MoO$_3$ (30%) system. As shown in Fig 11, the degradation percentage of TC after 120 min irradiation was 90% with free quencher and drastically decreased to about 6% when BQ was added into the system. In the same time, the addition of EDTA-2Na, K$_2$Cr$_2$O$_7$ and IPA resulted in 21, 39 and 83% photodegadation percentages of TC. These results indicated that

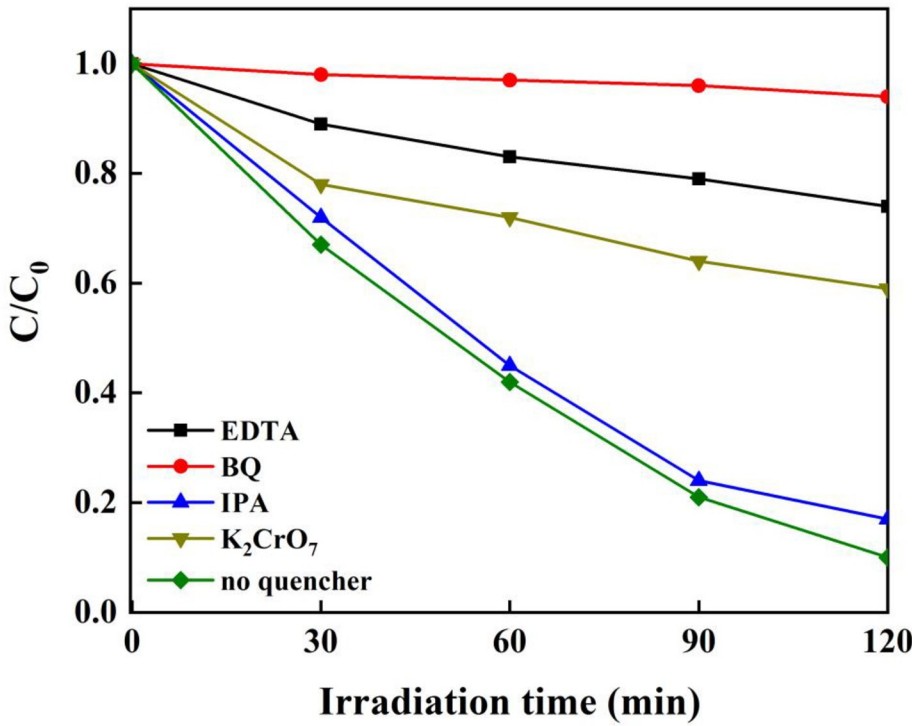

**Fig 11. Results of active species trapping experiments.**

$\cdot O_2^-$ played vital role for the TC photodegradation, $h^+$ and $e^-$ had modest contribution to the TC decomposition, while $\cdot OH$ was had the lowest contribution to the TC degradation. It should be noted that some other intermediates might be produced during the photodegradation reaction, which might take part in the degradation of TC.

In composites of two semiconductors, the effective separation of $e^-/h^+$ pairs depends on the appropriate band-gap positions of them. The band positions of g-C$_3$N$_4$ and MoO$_3$ could be obtained using empirical equations (Eqs 2 and 3) [23]:

$$E_{CB} = X - E_C - \frac{1}{2}E_g \qquad (2)$$

$$E_{VB} = E_{CB} + E_g \qquad (3)$$

Where $X$ is the absolute electronegativity of the atom semiconductor used to represent the geometric mean of the absolute electro-negativity of the constituent atoms, which is defined as the arithmetic mean of the atomic electron affinity and the first ionization energy; $E_{CB}$ is the energy of free electrons of the hydrogenscale (4.5 eV); $E_g$ is the band gap of the semiconductor; $E_{CB}$ is the conduction band potential and $E_{VB}$ is the valence band potential. According to previous studies, the absolute electronegativity $X$ for g-C$_3$N$_4$ and MoO$_3$ were 4.73 eV and 6.40 eV [65,66], respectively. From the Tauc's equation, $E_g$ of g-C$_3$N$_4$ and MoO$_3$ were to be 2.72 eV and 2.85 eV, respectively.

Based on the above analysis, the conduction bands (CB) of g-C$_3$N$_4$ and MoO$_3$ respectively were -1.13 and 0.47. Accordingly, the valance bands (VB) of them were 1.59 and 3.33, respectively. The results were similar to other studies [23]. Based on the results obtained by PL experiments, for the Fe$_3$O$_4$/g-C$_3$N$_4$/MoO$_3$ (30%) photocatalyst, the photogenerated $e^-$ and $h^+$ could

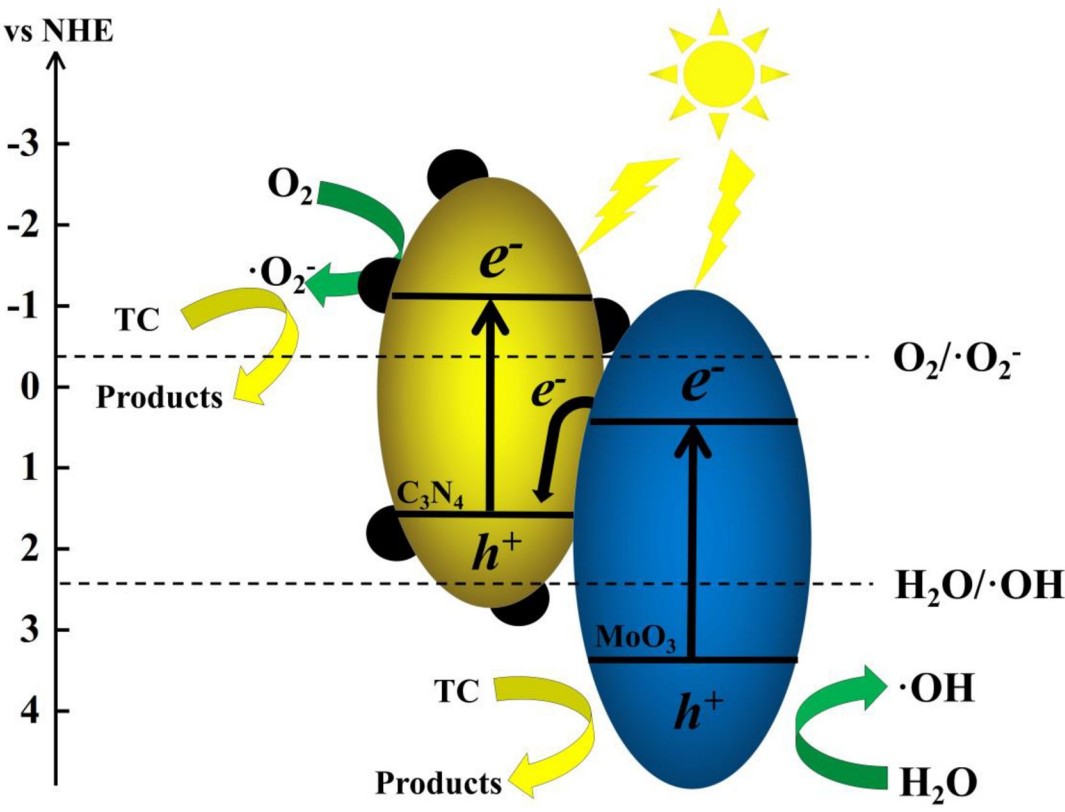

**Fig 12. Proposed mechanism for enhanced photocatalytic activity over Fe$_3$O$_4$/g-C$_3$N$_4$/MoO$_3$ nanocomposites.**

be effectively separated under visible light. According to the traditional mechanism, the e$^-$ in the CB of g-C$_3$N$_4$ could transfer to the CB of MoO$_3$ while the h$^+$ could migrate in the opposite direction. Generally, the reduction of O$_2$ with photoelectrons produced ·O$_2^-$ (e$^-$+ O$_2$→ ·O$_2^-$, O$_2$/·O$_2^-$ = -0.33 V vs. NHE) [67]. The ·OH$^-$ could be obtained by photoholes oxidized H$_2$O directly (h$^+$ + H$_2$O → ·OH + H$^+$, ·OH/OH$^-$ = 2.40 V vs. NHE) (Michael R. Hoffmann, 1995; Wen et al., 2017) or indirectly through ·O$_2^-$ (·O$_2^-$+ H$_2$O → H$_2$O$_2$ →·OH) [68]. In summary, the VB of MoO$_3$ and g-C$_3$N$_4$ are excited by visible light at the same time, and then the photo-electrons in the CB of MoO$_3$ and the holes in the solid-solid contact interface of the VB of g-C$_3$N$_4$ recombine, resulting in photoelectron retention In the CB of g-C$_3$N$_4$, holes are left in the VB of MoO$_3$. Therefore, g-C$_3$N$_4$ and MoO$_3$ could form Z-scheme and enhanced the separation of photogenetrated e$^-$/h$^+$ pairs at the interface of Fe$_3$O$_4$/g-C$_3$N$_4$/MoO$_3$ [64]. As shown in Fig 12, under light irradiation, the e$^-$ in the CB of g-C$_3$N$_4$ had relative stability thus benefited to the continuous generation of O$_2^-$ from O$_2$. The h$^+$ in the VB of MoO$_3$ generating ·OH$^-$ by oxi-dized H$_2$O. Some of h$^+$ in the VB of MoO$_3$ took part in the oxidation of TC, while the rest h$^+$ were reduce H$_2$O to·OH, which was not the main reactive species for TC degradation.

## 4. Conclusions

In this study, a novel and easily separated ternary Fe$_3$O$_4$/g-C$_3$N$_4$/MoO$_3$ (30%) photocatalyst was presented using melamine, FeCl$_3$, FeCl$_2$, and AHM as materials. This catalyst provided enhanced photocatalytic activity toward the removal of TC in aqueous environment. The photocatalytic activity of the novel catalyst was approximately 6.9 times of MoO$_3$, 5 times of g-C$_3$N$_4$, and 19.9 times of Fe$_3$O$_4$/g-C$_3$N$_4$ on photodegradation of TC. The excellent

photodegrading ability was due to the formation of Z-scheme structure between $C_3N_4$ and $MoO_3$, which could effectively separate the photogenerated $e^-/h^+$ pairs and efficiently utilize the $e^-$ and $h^+$. The highly improved TC-photodegrading ability was also beneficial from the wide range light absorption. This work indicated that the novel $Fe_3O_4/g\text{-}C_3N_4/MoO_3$ was beneficial in decreasing TC and other environmental pollutants with high-level concentration in water, and paved a new way to the development of photocatalytic technology.

## Supporting information

**S1 Fig. Schematic diagram of photocatalytic reaction device.**
(DOC)

**S2 Fig. TGA curves of pure $g\text{-}C_3N_4$ and $Fe_3O_4/g\text{-}C_3N_4/MoO_3$ (30%) photocatalysts.**
(DOC)

**S1 Table. Comparison of degradation performance of similar photocatalysts.**
(DOC)

## Author Contributions

**Conceptualization:** Tianpei He.

**Data curation:** Yaohui Wu, Chenyang Jiang, Zhifen Chen.

**Formal analysis:** Tianpei He, Xiaoyong Chen.

**Writing – original draft:** Tianpei He.

**Writing – review & editing:** Yonghong Wang, Gaoqiang Liu, Zhenggang Xu, Ge Ning, Yunlin Zhao.

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
