## [Decision Letter · Decision Letter 0]

31 Mar 2020

PONE-D-20-07918

Title - Novel magnetic Fe3O4/g-C3N4/MoO3 nanocomposites with highly enhanced photocatalytic activities: visible-light-driven degradation of tetracycline from aqueous environment

PLOS ONE

Dear Dr. Yaohui Wu,

Thank you for submitting your manuscript to PLOS ONE. After careful consideration, we feel that it has merit but does not fully meet PLOS ONE’s publication criteria as it currently stands. Therefore, we invite you to submit a revised version of the manuscript that addresses the points raised during the review process.

The article reports on the synthesis of Fe3O4/g-C3N4/MoO3 photocatalytic material and its application towards TC degradation.  However, this work would be a valuable addition to the field and enrich the readership. But the manuscript in its present form needs to address some critical technical points raised by reviewers, therefore, I recommend the publication of this manuscript in Journal PLOS ONE after addressing some technical aspects, grammatical improvement.

We would appreciate receiving your revised manuscript by May 15 2020 11:59PM. To enhance the reproducibility of your results, we recommend that if applicable you deposit your laboratory protocols in protocols.io, where a protocol can be assigned its own identifier (DOI) such that it can be cited independently in the future. For instructions see: http://journals.plos.org/plosone/s/submission-guidelines#loc-laboratory-protocols

We look forward to receiving your revised manuscript.

Kind regards,

Satya Pal Nehra, PhD

Academic Editor

PLOS ONE

Additional Editor Comments (if provided):

The article reports on the synthesis of Fe3O4/g-C3N4/MoO3 photocatalytic material and its application towards TC degradation. However, this work would be a valuable addition to the field and enrich the readership. But the manuscript in its present form needs to address some critical technical points raised by reviewers, therefore, I recommend the publication of this manuscript in Journal PLOS ONE after addressing some technical aspects, grammatical improvement.

Journal Requirements:

2. Please ensure that in your methods section you have provided details of the sources of all materials, chemicals, equipment and instrumentation used in your study, including manufacturer/names. This is in line with our reproducibility criterion for publishing, see https://journals.plos.org/plosone/s/criteria-for-publication#loc-3

5. Please include your tables as part of your main manuscript and remove the individual files. Please note that supplementary tables (should remain/ be uploaded) as separate "supporting information" files

6. We note you have included a table to which you do not refer in the text of your manuscript. Please ensure that you refer to Table 1-2 in your text; if accepted, production will need this reference to link the reader to the Table.

Reviewers' comments:

Reviewer's Responses to Questions

**Comments to the Author**

1. Is the manuscript technically sound, and do the data support the conclusions?

Reviewer #1: Yes

Reviewer #2: Partly

2. Has the statistical analysis been performed appropriately and rigorously? 

Reviewer #1: Yes

Reviewer #2: Yes

3. Have the authors made all data underlying the findings in their manuscript fully available?

Reviewer #1: Yes

Reviewer #2: Yes

4. Is the manuscript presented in an intelligible fashion and written in standard English?

Reviewer #1: Yes

Reviewer #2: Yes

5. Review Comments to the Author

Reviewer #1: In this paper, the authors explored the Novel magnetic Fe3O4/g-C3N4/MoO3 nanocomposites with highly enhanced photocatalytic activities: visible-light-driven degradation of tetracycline from aqueous environment. The overall idea is relatively clear and the logic is relatively reasonable. However, some questions need be seriously addressed to reach the high standard for publishing. The detailed comments are listed as follows.

1. There are some typo errors, for example; space between two words/ two sentences. I would suggest to go through the

manuscript again to remove these typo errors.

2. UV-vis word in abstract should be replaced with UV-Vis and check it throughout the manuscript.

3. The light intensity should be provided when conducting photocatalytic experiments.

4. There are many literature available with similar kind of work and results. All the adsorbents reported be compared in a

tabular form to establish the superiority of the work.

5. The quality of figures 2,3 and 9 should be enhanced. However, the scale bar is not clear in figure 2, so please redraw

the figure. In result and discussion ,paragraph second if fig 2 (d, e and f) is Fe,C and N then what is fig 2(b and c). Moreover fig 2 (b,c,d,e

and f) is not mentioned anywhere in the manuscript.

6. The advantages of using composites, especially heterostructures need be introduced and the following papers need be

cited: Synthesis, characterization and application of silver doped graphitic carbon nitride as photocatalyst towards visible

light photocatalytic hydrogen evolution, International Journal of Hydrogen Energy(2019), Effect of Calcination

Temperature, pH and Catalyst Loading on Photodegradation Efficiency of Urea Derived Graphitic Carbon Nitride towards

Methylene Blue Dye Solution, Royal Society of Chemistry(2019), 9(15381-15391). ZnO-Modified g-C3N4: A Potential

Photocatalyst for Environmental Application, American Chemical Society(2020). Biogenic mediated Ag/ZnO

nanocomposites for photocatalytic and antibacterial activities towards disinfection of water, Journal of Colloid and

Interface Science, (2020),563(370-380), Phytoextract mediated ZnO/MgO nanocomposites for photocatalytic and

antibacterial activities, Journal of Photochemistry and Photobiology A: Chemistry(2019),385(112049).

Reviewer #2: Manuscript Number: PONE-D-20-07918

Manuscript Title: Novel magnetic Fe3O4/g-C3N4/MoO3 nanocomposites with highly enhanced photocatalytic activities: visible-light-driven degradation of tetracycline from aqueous environment

The article reports on the synthesis of Fe3O4/g-C3N4/MoO3 photocatalytic material and its application towards TC degradation. There are reports on graphitic carbon nitride and it combined with other systems, and their applications in photocatalysis, sensor technology, water splitting and so on. However, this work would be a valuable addition to the field and enrich the readership. But the manuscript in its present form needs to address some critical technical points, therefore, I recommend the publication of this manuscript in Journal PLOS ONE after addressing some technical aspects, grammatical and typographical errors.

1. Update references with recent examples describing the use of visible light responding g-C3N4 based photocatalysts (RSC Advances, 9 (2019), 15381-15391; Journal of Nanoscience and Nanotechnology, 19 (8) (2019), 5241-5248; Physica E: Low-dimensional Systems and Nanostructures, 114 (2019), 113560; International Journal of Hydrogen Energy, doi:10.1016/j.ijhydene.2019.06.06. ; ACS Omega 2020, 5, 8, 3828-3838)

2. There are various typo errors throughout the manuscript, for instance in line no. 4 “,” should be removed after MoO3, line no. 20 “.” should be checked for pg no. 8 and similarly for whole manuscript.

3. Author has reported the effect of weight percent of MoO3 present in the composite but no evidence has been provided to confirm the composite elaborated weight composition of final ternary composite, therefore TGA should be done to support the stated fact.

4. Author should mention the complete details of photocatalytic instrument taken in use for the degradation studies like its capacity, the distance of light source from the surface of pollutant and light source intensity etc.

5. XRD patterns graph should be expressed more clearly, the peaks of g-C3N4 in 10% and g-C3N4/Fe3O4 composite is not clearly visible in present graph style.

6. Authors must explain that how the compound weight percentage has been calculated from EDX data? EDX analysis can only give estimation for elemental composition and not compound in particular. The EDX analysis needs reconsideration.

7. Fig 2(b,C) should also be represented with corresponding elements.

8. As mentioned “Pure g-C3N4 (Fig. 3(b)) consisted of a transparent sheet structure” is not appearing as explained in TEM image. In image 3b the sheet like structure is not visible therefore need explanation or replacement.

9. The language should be improved throughout the manuscript.

6. PLOS authors have the option to publish the peer review history of their article (what does this mean?). If published, this will include your full peer review and any attached files.

Reviewer #1: No

Reviewer #2: No

---

## [Author Response · Author response to Decision Letter 0]

22 Jul 2020

Responses to Reviewers’ Comments

Dear Editor and Reviewers:

Thank you very much for making a scrutiny into our manuscript. We appreciate editor and reviewers very much for their positive and constructive comments and suggestions on our manuscript which are all valuable and helpful for revising and improving our paper. The manuscript has been carefully revised according to the comments and suggestions, the revised parts are marked in red. Responses to the comments and suggestions are listed below. We sincerely hope the modification can conform to the request of yours. We sincerely hope this manuscript will be finally acceptable to be published on Plos One. Thank you very much for all your help and looking forward to hearing from you soon. 

Reviewer #1: 

Comment 1: 

-There are some typo errors, for example; space between two words/ two sentences. I would suggest to go through the manuscript again to remove these typo errors.

Response:

Thanks for your indication and suggestion.After reading the manuscript many times, these typing errors have been further corrected. We are very sorry for this negligence.

Comment 2: 

-UV-vis word in abstract should be replaced with UV-Vis and check it throughout the manuscript.

Response:

Thanks for your indication and suggestion.After many inspections, We have re-checked the full text to eliminate this error, which has been corrected and marked in red. A total of three errors were found, including line 11 on page 1, line 11 on page 17, and line 7 on page 33. Sorry for this negligence.

Comment 3: 

-The light intensity should be provided when conducting photocatalytic experiments.

Response: 

Thank you for your kind suggestion. Based on your kind suggestions, we detected the light intensity in the photocatalytic experiment, and the result was displayed in line 1-5 page 7 in the MS, which expressed as: The capacity of the synthesized catalysts to photodegrade TC was performed by a photochemistry reaction instrument (YM-GHX-V, Shanghai Yuming Instrument Co. Ltd, China) with a 1000 W Xe lamp applied as visible light source, as shown in Fig. S1. In the reaction system, the reaction solution is packed in a quartz tube with a capacity of 50 ml, and the quartz tube is fixed at a distance of 2 cm from the light source. An optical power meter (OPT-1A, China) was used to measure the intensity of the experimental lamp to be 37.5 mW/cm2 (λ >400 nm). A water circulation system was utilized to keep the reaction system at 15 oC. In each experiment, 10 mg of the photocatalyst was added into 50 mL of TC solution (40 mg/L).

Fig. S1 Schematic diagram of photocatalytic reaction device.

Comment 4: 

-There are many literature available with similar kind of work and results. All the adsorbents reported be compared in a tabular form to establish the superiority of the work.

Response:

Thanks for your indication and suggestion. According to recommendations, We compared the photocatalytic performance of the photocatalyst in this study with some heterostructure photocatalysts in other studies to illustrate the advantages of this work. It is mentioned atline 14 page 17 in the MS and the results are listed in Table S1, which written as the following: Comparison of Fe3O4/g-C3N4/MoO3 (30%) with other similar reported systems of Fe3O4/g-C3N4 composites has been discussed in Table S1.

S.

No. Photocatalysts Source of

illumination Cphotocatlyst

(mg mL−1) Cpollutant 

(mg L−1) Time

(min) Pollutant Photocatalytic degradation

efficiency (%) Refs.

1 g-C3N4/MoO3(7%) 300W Xenon lamp 1 10 180 MB 93 23

2 1.5 wt% MoO3-C3N4 350W Xenon lamp 1 20 120 MO 87 30

3 CNFO-15.2 300 W Xenon lamp 0.25 5 60 RhB 97 35

4 Fe3O4/g-C3N4/MoO3(30%) 1000W

Xenon lamp 0.2 40 120 TC 94 This work

Tab. S1 Comparison of degradation performance of similar photocatalysts.

Comment 5: 

-The quality of figures 2, 3 and 9 should be enhanced. However, the scale bar is not clear in figure 2, so please redraw the figure. In result and discussion ,paragraph second if fig 2 (d, e and f) is Fe,C and N then what is fig 2(b and c). Moreover fig 2 (b,c,d,e and f) is not mentioned anywhere in the manuscript.

Response: 

Thank you for your kind suggestion. Based on your suggestions, we have redrawn figures 2, 3 and 9, and re-explained and accurately explained and analyzed Fig 2 (d, e, and f) in the 5 line of the 10 page of the manuscript, which expressed as: It could be clearly found C, N, Fe, O and Mo (Fig. 2(b-f)) were all homogeneous indicating uniform distributions of Fe3O4, g-C3N4, and MoO3 in the selected area of the corresponding SEM image (Fig. 2a).

Comment 6:

-The advantages of using composites, especially heterostructures need be introduced and the following papers need be cited: Synthesis, characterization and application of silver doped graphitic carbon nitride as photocatalyst towards visible light photocatalytic hydrogen evolution, International Journal of Hydrogen Energy(2019), Effect of Calcination Temperature, pH and Catalyst Loading on Photodegradation Efficiency of Urea Derived Graphitic Carbon Nitride towards Methylene Blue Dye Solution, Royal Society of Chemistry(2019), 9(15381-15391). ZnO-Modified g-C3N4: A Potential Photocatalyst for Environmental Application, American Chemical Society(2020). Biogenic mediated Ag/ZnO nanocomposites for photocatalytic and antibacterial activities towards disinfection of water, Journal of Colloid and Interface Science, (2020),563(370-380), Phytoextract mediated ZnO/MgO nanocomposites for photocatalytic and antibacterial activities, Journal of Photochemistry and Photobiology A: Chemistry(2019),385(112049).

Response: 

Thanks for the reviewer’s good evaluation and kind suggestion. We read these papers carefully and found their works were helpful for ours, so we have cited them as reference as [8], [9], [13], [17], [22]. The revised section was marked as red letters in lines 6, 14, and 17 on page 3, lines 1 on page 4, which expressed as: Some photocatalysts have the function of degrading pollutants while Excellent antibacterial activity [8-9]. Due to its advantages of low toxicity, low preparation cost and high stability, it has been applied to the removal of organic pollutants in water, which has aroused extensive research interest [13-14]. Therefore, various methods have been evolved to enhance the photocatalytic activity of pure g-C3N4, including metal deposition [17-18]. By coupling g-C3N4 with other semiconductors to form a heterojunction structure, the shortcomings of high recombination rate of photogenerated electron-hole pairs of a single photocatalyst could be solved [22].

Reviewer #2: 

Comment 1: 

-Update references with recent examples describing the use of visible light responding g-C3N4 based photocatalysts (RSC Advances, 9 (2019), 15381-15391; Journal of Nanoscience and Nanotechnology, 19 (8) (2019), 5241-5248; Physica E: Low-dimensional Systems and Nanostructures, 114 (2019), 113560; International Journal of Hydrogen Energy, doi:10.1016/j.ijhydene.2019.06.06.; ACS Omega 2020, 5, 8, 3828-3838) Royal Society of Chemistry(2019), 9(15381-15391). ZnO-Modified g-C3N4: A Potential Photocatalyst for Environmental Application, American Chemical Society(2020). Biogenic mediated Ag/ZnO nanocomposites for photocatalytic and antibacterial activities towards disinfection of water, Journal of Colloid and Interface Science, (2020),563(370-380), Phytoextract mediated ZnO/MgO nanocomposites for photocatalytic and antibacterial activities, Journal of Photochemistry and Photobiology A: Chemistry(2019),385(112049).

Response:

Thanks for the reviewer’s good indication and suggestion. Based on your comments, we read these papers carefully and found their works were helpful for ours, so we have cited them as reference as [8], [9], [13], [14], [17], [20], [22]. The revised content is located on the page 3 line 6, 14, 20, 22 and page 4 line 1 and the revised part has been marked in red, which expressed as: Some photocatalysts have the function of degrading pollutants while Excellent antibacterial activity [8-9]. Due to its advantages of low toxicity, low preparation cost and high stability, it has been applied to the removal of organic pollutants in water, which has aroused extensive research interest [13-14]. Therefore, various methods have been evolved to enhance the photocatalytic activity of pure g-C3N4, including metal deposition [17-18], nonmetal doping [19], coupling with other materials [20], and using nano-sized structures [21]. By coupling g-C3N4 with other semiconductors to form a heterojunction structure, the shortcomings of high recombination rate of photogenerated electron-hole pairs of a single photocatalyst could be solved [22]. Unfortunately, one of the papers you recommended has not been found after our multiple searches (International Journal of Hydrogen Energy, doi:10.1016/j.ijhydene.2019.06.06.). Please re-confirm whether the corresponding journal or doi is correct.

Comment 2: 

-There are various typo errors throughout the manuscript, for instance in line no. 4 “,” should be removed after MoO3, line no. 20 “.” should be checked for pg no. 8 and similarly for whole manuscript.

Response: 

Thanks for your indication and suggestion. We apologize for our negligence, we have checked many times to ensure that similar mistakes will not occur.

Comment 3:

-Author has reported the effect of weight percent of MoO3 present in the composite but no evidence has been provided to confirm the composite elaborated weight composition of final ternary composite, therefore TGA should be done to support the stated fact.

Response: 

Thanks for the reviewer’s good evaluation and kind suggestion. Based on your opinion, the TGA experiment is supplemented in our revised MS, and the weight percentage of each component in the composite material is calculated and analyzed. We rewritten the relevant content as: Thermo-gravimetric analysis (TGA) was carried out on a STA 449F3 thermal analyzer with a heating rate of 10 oC/min from room temperature to 1000 oC in an air flow. Fig. S2 displays TGA curves for the g-C3N4 and Fe3O4/g-C3N4/MoO3 (30%) samples. As can be seen, the pristine g-C3N4 shows a weight loss of about 96% after heating up to 750 oC. Hence, it was concluded that the g-C3N4 decomposes almost completely heating up to 750 oC. It is evident that the thermal behavior of Fe3O4/g-C3N4 and Fe3O4/g-C3N4/MoO3 (30%) samples are similar to that of g-C3N4. As can be seen, by loading Fe3O4 and MoO3 on the g-C3N4 sheets, thermal degradation of the nano-composites starts from lower temperature relative to the pristine g-C3N4. Hence, similar to many g-C3N4-based nanocomposites, thermal stability of the pristine g-C3N4 decreases with depositing different particles [45-46]. The g-C3N4 contents of Fe3O4/g-C3N4 and Fe3O4/g-C3N4/MoO3 (30%) nanocomposites were calculated from the weights remaining after heating the samples to over 650 oC. The g-C3N4 contents of the Fe3O4/g-C3N4/MoO3 (30%) nanocomposite was about 8.2%, respectively. As can be seen, besides the weight loss of g-C3N4, another weight loss between 750 and 1000 oC in the Fe3O4/g-C3N4/MoO3 (30%) composites, could be ascribed to the vaporization of MoO3. The MoO3 contents of the Fe3O4/g-C3N4/MoO3 (30%) is about 16.4 %. In addition, after calculation, The MoO3 contents of the Fe3O4/g-C3N4/MoO3 (30%) is about 75.4 %. The results were listed in Tab. 1, which were marked in red from page 7 line 17, page 12 line 21,The results are shown in Fig. S2 and Tab. 1.

Fig. S2 TGA curves of pure g-C3N4 and Fe3O4/g-C3N4/MoO3 (30%) photocatalysts.

Compound Weight percentage

g-C3N4 8.2

MoO3 16.4

Fe3O4 75.4

Tab. 1. Weight percentages of different compounds in the Fe3O4/g-C3N4/MoO3 (30%)nanocomposite.

Comment 4:

-Author should mention the complete details of photocatalytic instrument taken in use for the degradation studies like its capacity, the distance of light source from the surface of pollutant and light source intensity etc.

Response: 

Thanks for the reviewer’s good evaluation and kind suggestion. According to your opinion, the complete details of the photocatalytic instrument for degradation research, such as its capacity, the distance between the light source and the surface of the pollutant, and the intensity of the light source are supplemented in our photocatalytic experimental design section. The result was displayed in line 1 page 8 in the MS, which expressed as: The capacity of the synthesized catalysts to photodegrade TC was performed by a photochemistry reaction instrument (YM-GHX-V, Shanghai Yuming Instrument Co. Ltd, China) with a 1000 W Xe lamp applied as visible light source, as shown in Fig. S1. In the reaction system, the reaction solution is packed in a quartz tube with a capacity of 50 ml, and the quartz tube is fixed at a distance of 2 cm from the light source. An optical power meter (OPT-1A, China) was used to measure the intensity of the experimental lamp to be 37.5 mW/cm2 (λ >400 nm). A water circulation system was utilized to keep the reaction system at 15 oC. In each experiment, 10 mg of the photocatalyst was added into 50 mL of TC solution (40 mg/L).

Fig. S1 Schematic diagram of photocatalytic reaction device.

Comment 5:

-XRD patterns graph should be expressed more clearly, the peaks of g-C3N4 in 10% and g-C3N4/Fe3O4 composite is not clearly visible in present graph style.

Response: 

Thanks for your indication and suggestion. According to your suggestion, in order to clearly express the XRD spectrum, we have redrawn the XRD spectrum, in which the peaks of g-C3N4 in 10% and g-C3N4/Fe3O4 composite is obvious.

Fig. 1. XRD patterns for the MoO3, g-C3N4, Fe3O4, Fe3O4/g-C3N4 and Fe3O4/g-C3N4/MoO3 nanocomposites.Inset image shows the deconvulation peaks for MoO3 and g-C3N4.

Comment 6:

-Authors must explain that how the compound weight percentage has been calculated from EDX data? EDX analysis can only give estimation for elemental composition and not compound in particular. The EDX analysis needs reconsideration.

Response: 

Thank you for your kind suggestion. Based on your suggestions, we performed the TGA experiment, and re-evaluated the mass ratio of each compound based on TGA analysis. The results were list in Table 1. The EDX analysis were applied to display the elemental distributions. The related content was listed in line 6 page 10.

Compound Weight percentage

g-C3N4 8.2

MoO3 16.4

Fe3O4 75.4

Tab. 1. Weight percentages of different compounds in the Fe3O4/g-C3N4/MoO3 (30%)nanocomposite.

Comment 7:

-Fig 2(b,C) should also be represented with corresponding elements.

Response:

Thanks for your indication and suggestion. Based on your suggestions, we have re-marked Fig 2. We are very sorry for such mistakes.

Fig. 2. (a) SEM images of Fe3O4/g-C3N4/MoO3 (30%); (b-f) EDX mapping for the Fe3O4/g-C3N4/MoO3 (30%) nanocomposite.

Comment 8:

-As mentioned “Pure g-C3N4 (Fig. 3(b)) consisted of a transparent sheet structure” is not appearing as explained in TEM image. In image 3b the sheet like structure is not visible therefore need explanation or replacement.

Response:

Thanks for the reviewer’s good evaluation and kind suggestion. Based on your suggestions, we have replaced the “transparent sheet structure” in the analysis of the TEM image with “lamellar-like and smooth morphology”. This modified part is located on line 10 of page 17, and has been marked in red. 

Comment 9:

The language should be improved throughout the manuscript.

Response:

Thank you for your kind suggestion. Based on your opinion, we have further improved the fluency of the sentences in the manuscript to make it easier to read and understand.

Thanks very much again for your comments and suggestions to our manuscript. We have tried our best to improve the manuscript and made some changes in the manuscript. We sincerely hope the revised manuscript can meet the requirements of yours.

Yours sincerely, 

Yaohui Wu

---

## [Editor Report · Decision Letter 1]

27 Jul 2020

Title - Novel magnetic Fe3O4/g-C3N4/MoO3 nanocomposites with highly enhanced photocatalytic activities: visible-light-driven degradation of tetracycline from aqueous environment

PONE-D-20-07918R1

Dear Dr. Wu,

We’re pleased to inform you that your manuscript has been judged scientifically suitable for publication and will be formally accepted for publication once it meets all outstanding technical requirements.

Kind regards,

Satya Pal Nehra, PhD

Academic Editor

PLOS ONE

Additional Editor Comments (optional):

Authors have answered each and every query raised by reviewers. Now the revised version of the manuscript can be accepted for the publication.
---

## [Editor Report · Acceptance letter]

3 Aug 2020

PONE-D-20-07918R1 

Novel magnetic Fe3O4/g-C3N4/MoO3 nanocomposites with highly enhanced photocatalytic activities: visible-light-driven degradation of tetracycline from aqueous environment 

Dear Dr. Wu:

I'm pleased to inform you that your manuscript has been deemed suitable for publication in PLOS ONE. Congratulations! Your manuscript is now with our production department. 

Kind regards, 

on behalf of

Dr. Satya Pal Nehra 

Academic Editor

PLOS ONE